# CDK4/6 or MAPK blockade enhances efficacy of EGFR inhibition in oesophageal squamous cell carcinoma

Jin Zhou[1],[*], Zhong Wu[1],[*], Gabrielle Wong[1], Eirini Pectasides[1], Ankur Nagaraja[1], Matthew Stachler[1], Haikuo Zhang[1], Ting Chen[1], Haisheng Zhang[1], Jie Bin Liu[1], Xinsen Xu[1], Ewa Sicinska[1],[2], Francisco Sanchez-Vega[3], Anil K. Rustgi[4], J. Alan Diehl[5], Kwok-Kin Wong[1],[6] & Adam J. Bass[1],[6],[7],[8]

Oesophageal squamous cell carcinoma is a deadly disease where systemic therapy has relied upon empiric chemotherapy despite the presence of genomic alterations pointing to candidate therapeutic targets, including recurrent amplification of the gene encoding receptor tyrosine kinase epidermal growth factor receptor (EGFR). Here, we demonstrate that EGFR-targeting small-molecule inhibitors have efficacy in EGFR-amplified oesophageal squamous cell carcinoma (ESCC), but may become quickly ineffective. Resistance can occur following the emergence of epithelial–mesenchymal transition and by reactivation of the mitogen-activated protein kinase (MAPK) pathway following EGFR blockade. We demonstrate that blockade of this rebound activation with MEK (mitogen-activated protein kinase kinase) inhibition enhances EGFR inhibitor-induced apoptosis and cell cycle arrest, and delays resistance to EGFR monotherapy. Furthermore, genomic profiling shows that cell cycle regulators are altered in the majority of EGFR-amplified tumours and a combination of cyclin-dependent kinase 4/6 (CDK4/6) and EGFR inhibitors prevents the emergence of resistance in vitro and in vivo. These data suggest that upfront combination strategies targeting EGFR amplification, guided by adaptive pathway reactivation or by co-occurring genomic alterations, should be tested clinically.

[1] Department of Medical Oncology, Dana-Farber Cancer Institute, Boston, Massachusetts 02215, USA. [2] Department of Molecular Oncologic Pathology, Dana-Farber Cancer Institute, Boston, Massachusetts 02215, USA. [3] Division of Computational Biology, Memorial Sloan-Kettering Cancer Center, New York, New York 10065, USA. [4] Department of Medicine, University of Pennsylvania School of Medicine, Philadelphia, Pennsylvania 19104, USA. [5] Department of Biochemistry and Molecular Biology, Hollings Cancer Center, The Medical University of South Carolina, Charleston, South Carolina 29425, USA. [6] Department of Medicine, Harvard Medical School, Boston, Massachusetts 02215, USA. [7] Cancer Program, The Broad Institute of MIT and Harvard, Cambridge, Massachusetts 02139, USA. [8] Department of Medicine, Brigham and Women's Hospital, Boston, Massachusetts 02115, USA. * These authors contributed equally to this work. Correspondence and requests for materials should be addressed to A.J.B. (email: adam_bass@dfci.harvard.edu).

Oesophageal cancer (EC) is the sixth leading cause of cancer mortality worldwide[1]. The most common variant of the disease is oesophageal squamous cell carcinoma (ESCC) that is highly prevalent in Asia and the developing world and harbours a 5-year survival rate of <20% (refs 2,3). Current therapies are centred upon endoscopic resection, surgery and empiric chemotherapy and chemoradiotherapy, but these tumours are not routinely treated with biologic or targeted agents. Moreover, clinical development of such agents has largely not been driven by biomarkers such as the presence of somatic genomic alterations. However, the convergence of our expanding knowledge of the cancer genome and availability of targeted agents creates new opportunity for rational, biomarker-driven therapies for ESCC.

Genomic characterization in ESCC has demonstrated that 7–28% of tumours harbour amplifications of the gene encoding receptor tyrosine kinase epidermal growth factor receptor (EGFR)[4–6]. EGFR has proven an effective target in diseases spanning non-small-cell lung cancer[7] to colorectal adenocarcinoma[8], glioblastomas[9] and head and neck cancer[10]. In systematic preclinical cell line screening across lineages, EGFR-amplified ESCCs were among the 3% of lines with greatest sensitivity to EGFR inhibitor erlotinib[11]. Furthermore, prior clinical testing of EGFR small-molecule inhibitor gefitinib in unselected ESCC patients demonstrated the potential for efficacy in this population. The sole responder in a phase II trial was a patient with EGFR amplification, and patients with higher tumour EGFR expression showed significantly longer survival[12]. However, these clinical results also demonstrated that the clinical impact of monotherapy with EGFR-directed agents in ESCC, even with EGFR amplification, differs from the dramatic responses seen in EGFR-mutant lung cancer. Although these data support the potential utility of EGFR blockade, they also suggest the likely need to develop combination inhibitor strategies for EGFR in ESCC.

To develop enhanced therapeutic strategies in EGFR-amplified cancers, considerations include the basal dependence upon EGFR signalling, the likely duration of clinical response to EGFR blockade and predicted etiologies of resistance. Broadly, when targeted agents are introduced into the clinic, the patients who initially respond typically subsequently develop acquired resistance[13–17]. When such resistance is induced by a clear secondary mutation, such as a secondary EGFR T790M mutation in non-small-cell lung cancer, targeted use of an appropriate secondary inhibitor can be highly effective. In contrast, other aetiologies of resistance such as the emergence of epithelial–mesenchymal transition (EMT) may be more challenging to address once resistance has developed[18–21]. Accordingly, increasing emphasis has been placed upon the development of up-front combination regimens that may act to thwart resistance before it emerges, analogous to the use of combination antiretroviral therapies for treatment of the human immunodeficiency virus.

We therefore sought to further investigate in preclinical models the development of more effective strategies to target EGFR-amplified ESCCs. By addressing fundamental questions regarding the initial drug sensitivity of these models, the emergence of resistance and mechanisms of blocking resistance, we hope to speed our ability to bring optimal therapeutic strategies forward into clinical testing for these cancers.

## Results

**Amplified EGFR is a putative target in ESCC cell lines.** We first confirmed the status of EGFR as a putative amplified target in ESCC, evaluating data from The Cancer Genome Atlas, where we observed focal amplification of EGFR in 17% of cases (Fig. 1a). We next turned to an evaluation of the genomic copy number, as inferred by high-density single-nucleotide polymorphism arrays, and protein expression of EGFR in a panel of genetically defined ESCC cell line models. These results identified several ESCC cell lines, TE8, OE21, KYSE30, KYSE140, KYSE180, KYSE450 and KYSE520, with EGFR gene amplification[22,23]. Within these models, EGFR protein, EGFR phosphorylation and downstream effectors extracellular signal–regulated kinase (ERK) and AKT were variably present, but consistently higher than observed in two EGFR nonamplified ESCC lines, TE10 and KYSE70 (Fig. 1b and Supplementary Fig. 1).

We next evaluated our panel of ESCC models for their in vitro sensitivity to erlotinib, a reversible small-molecule EGFR inhibitor, and afatinib, an irreversible small-molecule EGFR/ERBB2 inhibitor, finding a range of sensitivities (Fig. 1c and Supplementary Table 1). Among these cell lines, OE21, KYSE140 and KYSE450 had greater in vitro sensitivity to EGFR inhibitors. In contrast, TE8, KYSE30 and KYSE520 cell lines had substantially less growth inhibition. We therefore asked whether other genome alteration could impact the response of these models to erlotinib and afatinib. Available profiling of these lines through the Cancer Cell Line Encyclopedia effort found that KYSE450 harbours an EGFR mutation (S7681), and KYSE30 harbours an endogenous HRAS mutation at codon 61 (Q61L), providing rationale for the sensitivity and resistance in these lines, respectively (Supplementary Table 2). In contrast, TE8 and KYSE520 showed de novo resistance to EGFR inhibition, without any apparent genomic alterations. Evaluation of target engagement and biochemical effects of erlotinib and afatinib in these ESCC cell lines largely matched sensitivity data. EGFR phosphorylation was modestly blocked by 1 μM erlotinib and strongly blocked by 100 nM afatinib treatment in all cell lines, and the phosphorylation of AKT and ERK was clearly inhibited in the erlotinib/afatinib-sensitive lines OE21 and KYSE140. However, downstream signalling persisted or was only slightly inhibited by EGFR-directed kinase inhibitors in the resistant TE8, KYSE30 and KYSE520 cell lines (Fig. 1d and Supplementary Fig. 2).

We next sought to investigate the specific inhibition effects of erlotinib and afatinib on cell cycle progression after 48 h dosing, and apoptosis after 72 h dosing. Dramatic induction of G0–G1 cell cycle arrest and apoptosis were observed in EGFR-inhibited OE21 cells and KYSE140 cells (Fig. 1e,f). These data demonstrate the ability for erlotinib and afatinib to block downstream signalling pathway, inhibit proliferation and induce cell cycle arrest and apoptosis in selected EGFR-amplified ESCC cell lines.

**EMT mediates acquired resistance to EGFR inhibition.** Our results demonstrated clear potential for EGFR-directed therapy in EGFR-amplified ESCC but that sensitivity was not universal. To better assess possible mediators of failure of EGFR therapy, we next evaluated the highly drug-sensitive OE21 cell line and generated erlotinib-resistant versions by long-term culture in stepwise increases in drug concentration starting at 500 nM until the cells were able to proliferate in 5 μM erlotinib, 25 times the original half-maximal inhibitory concentration (IC$_{50}$; Fig. 2a). These cells were termed OE21ER and were confirmed via genotyping to originate from OE21 cells. We demonstrated that these OE21ER cells also were cross resistant to afatinib (Fig. 2a). We sought to investigate the biochemical changes accompanying resistance in the OE21ER cells. As shown in Fig. 2b, the basal level of phosphorylated EGFR was slightly lower in the resistant cells, and this is different than would be expected if the cells had acquired a secondary EGFR alteration making the drug no longer effective. However, OE21-resistant cells showed reactivation of

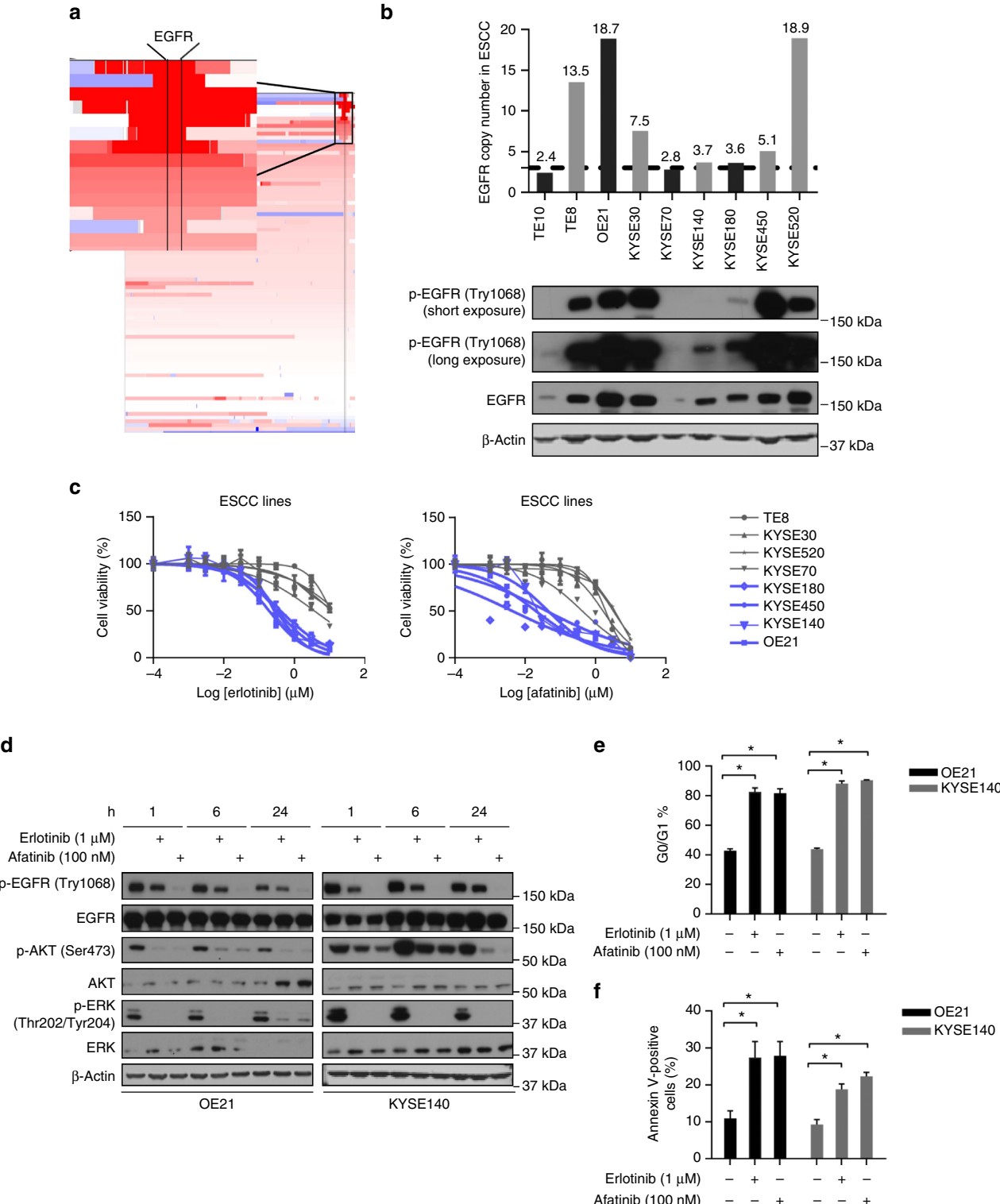

**Figure 1 | Amplified EGFR is a putative target in ESCC cell line models.** (**a**) Integrative Genomics Viewer (IGV) screenshots of chromosome 7p12.3-p12.1 and the EGFR locus in ESCC patients from The Cancer Genome Atlas (TCGA). The broader view shows chromosome 7p in 90 ESCC samples with the inset image focussed in at the EGFR locus in patients with copy-number gains. Red colour means copy-number gain and blue colour means copy-number loss (x axis: chromosomal coordinates; y axis: individual cases). (**b**) Single-nucleotide polymorphism (SNP) array inferred copy-number and immunoblots showing basal level of phosphorylation and total EGFR protein expression in a panel of ESCC cell line models and normal oesophageal squamous epithelial cell EPC. (**c**) Plots showing the *in vitro* sensitivity of a panel of ESCC cell line models to distinct EGFR inhibitors erlotinib and afatinib. Cell viability at distinct doses relative to vehicle-treated controls is shown. (**d**) Immunoblots evaluating the biochemical response to erlotinib and afatinib in representative EGFR inhibitor-sensitive cell line models. Cells were harvested at the indicated time points after treatment with 1 μM erlotinib or 100 nM afatinib. (**e**) Plots show analysis of cell cycle arrest after 48 h of inhibitor treatment with 1 μM erlotinib or 100 nM afatinib. (**f**) Plots show analysis of apoptosis after 72 h of treatment with 1 μM erlotinib or 100 nM afatinib. All experiments were performed in triplicate for each condition and repeated at least twice. All error bars represent s.d., $n \geq 3$. Student's *t*-test was used for statistical analysis. *$P < 0.05$.

downstream pathway as phospho-AKT and phospho-ERK. Moreover, afatinib inhibited EGFR phosphorylation in the resistant derivatives, but blockade of phosphorylated EGFR was

decoupled from inhibition of downstream signalling pathways, as evidenced by lack of inhibition of AKT and ERK phosphorylation.

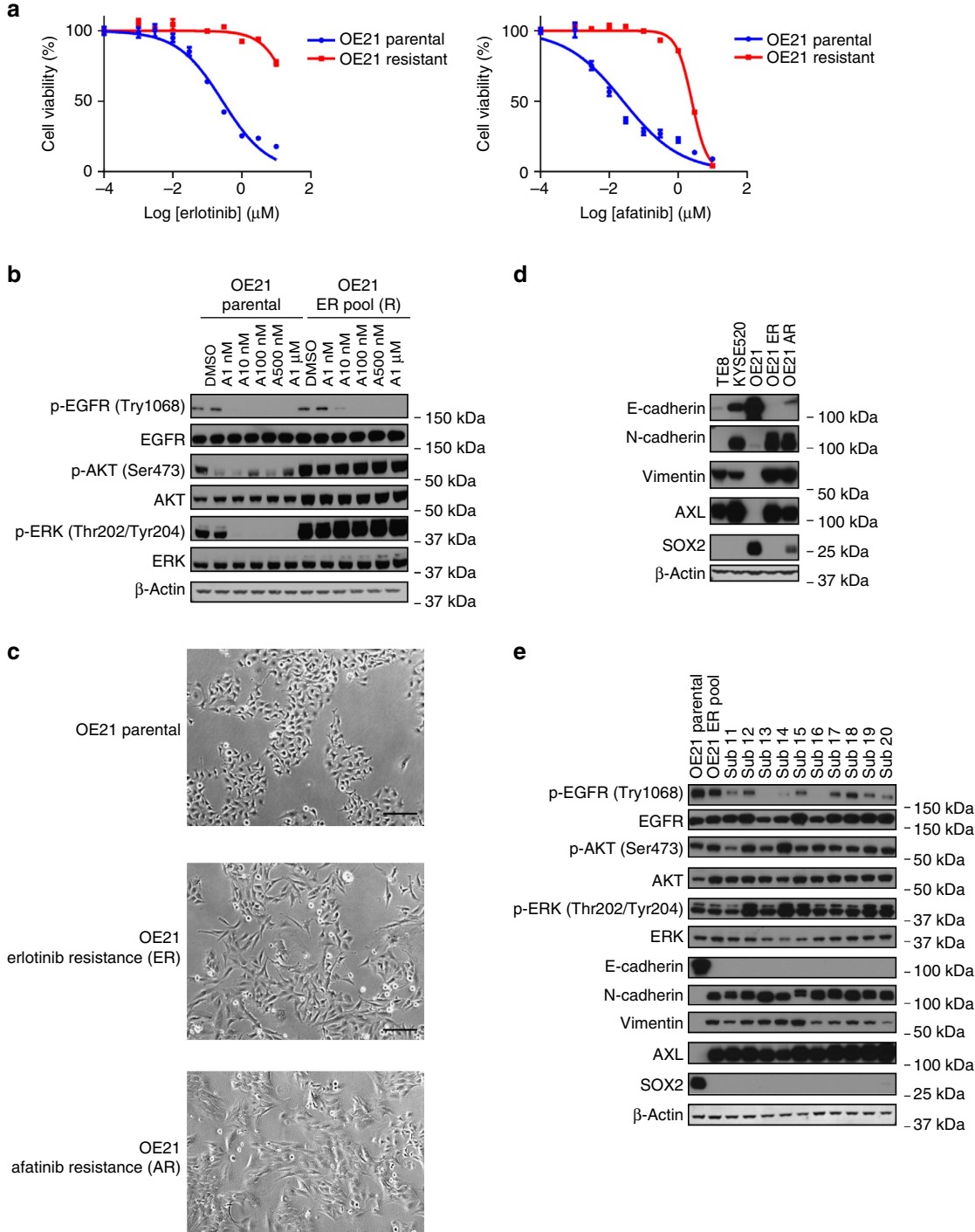

**Figure 2 | OE21 acquired resistance to EGFR inhibition *in vitro* with induction of an EMT phenotype.** (**a**) The EGFR-amplified ESCC OE21 cell-resistant variant was generated by the indicated culture with gradually increasing concentrations of erlotinib with resistance as confirmed by dose–response curve using Cell-Titer GLO. OE21 parental cells and erlotinib-resistant cells were treated with erlotinib and afatinib at indicated concentrations for 72 h and relative cell growth was quantified using the Cell-Titer-Glo assay and plotted as a percentage of growth relative to dimethylsulfoxide (DMSO)-treated control cells. Data points are represented as mean ± s.d. of three independent experiments. (**b**) Signalling responses of OE21 parental and resistant cells with increasing concentrations of afatinib. Cells were harvested 6 h after drug treatment. (**c**) Representative images of parental and erlotinib-resistant cells demonstrating apparent mesenchymal morphology in the resistant models. Scale bar, 100 μm. (**d**) Immunoblot measurement of candidate EMT markers of OE21 parental, erlotinib-resistant and afatinib-resistant cell lines, and TE8, KYSE520 cell lines. (**e**) Immunoblots evaluating distinct isolated OE21ER subclones for EGFR and downstream pathway phosphorylation and EMT marker expression. All experiments were performed in triplicate for each condition and repeated at least twice. All error bars represent s.d., *n* = 3.

Upon further inspection of the OE21ER cells, we also observed a morphologic change during their acquisition of resistance to erlotinib. They appeared to transition from their original densely packed adherent layer of small cells to a spindle-like morphology with loss of intercellular adhesion, increased intercellular separation and increased formation of pseudopodia (Fig. 2c), suggestive of an EMT transition, a phenomenon that has been observed in non-small-cell lung cancer in the context of acquired resistance to EGFR inhibitors[18]. To evaluate for EMT, we found that expression of the epithelial marker E-cadherin was decreased but that the mesenchymal marker N-cadherin and vimentin increased, consistent with an EMT phenotype (Fig. 2d). Furthermore, we found that resistant populations lose SOX2, a squamous lineage-dependent transcription factor, and upregulated AXL, a tyrosine kinase receptor, whose activity has been shown to promote resistance to EGFR-targeted therapy in lung cancer, in association with EMT transition (Fig. 2d)[24,25]. We also used a second EGFR inhibitor, afatinib, to generate resistant OE21 cell lines using the same protocol. Starting with a drug concentration 50 nM, we generated cells able to proliferate in 1 μM afatinib (OE21AR). These cells showed similar morphology change and molecular characteristics as OE21ER cells (Fig. 2c,d).

In order to further evaluate the mechanisms of resistance to EGFR inhibition, several individual resistant OE21ER subclones were isolated and confirmed to be drug resistant (Fig. 2e and Supplementary Fig. 3a). We noticed that the resistant cell population was heterogeneous and showed different morphology and growth rate (Supplementary Fig. 3b,c). Loss of E-cadherin and SOX2 expression, gain of N-cadherin, vimentin and AXL expression, however, were shown in all resistant subclones independently of the basal level of EGFR, AKT and ERK phosphorylation. (Fig. 2e and Supplementary Fig. 3a). We also generated KYSE140-erlotinib and KYSE140-afatinib resistant lines that are able to proliferate in 3 μM erlotinib and 300 nM afatinib. Those lines showed strong activation of phospho-ERK, and variable upregulation of vimentin and AXL (Supplementary Fig. 4).

Previous observations showed that cancer cell lines with pre-existing EMT have intrinsic resistance to EGFR inhibitors[23–28]. This possibility prompted us to evaluate for potential EMT in the EGFR+ ESCC lines that were at baseline insensitive to EGFR blockade. We specifically queried the presence of mesenchymal features and changes in E-cadherin, N-cadherin and vimentin expression. Indeed, TE8 had mesenchymal appearance, showed minimal E-cadherin but strong vimentin and AXL expression. In addition, KYSE520 showed strong expression of N-cadherin, vimentin and AXL. Both TE8 and KYSE520 also showed no expression of the squamous lineage marker SOX2 (Fig. 2d). These results suggest that a mesenchymal phenotype may lead to intrinsic resistance to EGFR blockade.

We next questioned the possible approaches to target OE21ER cells with the mesenchymal phenotype. We first sought to identify secondary targets whose blockade could augment the response to EGFR inhibition therapy. We tested the ability of mitogen-activated protein kinase kinase (MEK), phosphatidylinositol-3-kinase (PI3K), AKT or mammalian target of rapamycin inhibitors (Trametinib, GDC0941, MK2206 and rapamycin) to augment EGFR inhibitor sensitivity in OE21ER pool cells. These inhibitors did not have substantial effect in OE21ER cells despite adequate biochemical target engagement (Supplementary Fig. 5a–d), suggesting the difficulty of treating these tumours upon the emergence of EMT. We also sought to test whether AXL could serve as an effective target for overcoming acquired resistance to EGFR inhibition. In the OE21ER cells, we first evaluated R428, a pharmacological small-molecule ATP competitor of AXL in

combination with EGFR inhibitors. R428 enhanced the antiproliferative activity of EMT-associated intrinsic and acquired resistant cell lines other than one RAS mutant line (Supplementary Fig. 6a–f), but was not sufficient to achieve greater inhibition of downstream pathways. Moreover, genomically silencing AXL could not resensitize these resistance cells to EGFR inhibition or block downstream pathways (Supplementary Fig. 7a–e), suggesting that AXL may not serve as an effective target to sensitize mesenchymal EGFR-amplified ESCC to EGFR inhibition. Together, the failure to readily overcome EGFR inhibitor resistance signifies the importance of identifying means of inhibiting the initial development of resistance in these tumours with effective up-front combination therapy.

**MAPK reactivation mediates resistance to EGFR inhibition.** We therefore next sought to investigate possible combination treatment approaches that may thwart the initial emergence of resistance in EGFR-amplified ESCC. We first asked what adaptive biochemical response exists in ESCC following EGFR inhibitor treatment. Biochemical time-course analysis in OE21 cells showed that although ERK1/2 and AKT were initially inhibited to similar degrees, ERK1/2 reactivation was observed within just 48 h following continuous exposure to treatment with erlotinib or afatinib, despite a much weaker level of reactivation of the phosphorylation of AKT and EGFR (Fig. 3a). The other EGFR inhibitor-sensitive lines showed similar ERK reactivation (Supplementary Fig. 8). These data, consistent with recent results seen in EGFR-mutant lung cancer[29], suggested the hypothesis that ERK rebound could compensate for EGFR blockade. Next, we assessed whether the addition of the MEK inhibitor trametinib to erlotinib or afatinib was able to block ERK reactivation. We observed that even with a dose of only 2 nM of trametinib, MEK blockade could inhibit rebound ERK1/2 phosphorylation (Fig. 3b).

Given these results, we next investigated whether concomitant MEK and EGFR blockade could enhance both the initial efficacy of therapy and the durability of response to EGFR blockade. As shown in Fig. 3c, single-agent trametinib showed minimal inhibition of the cell proliferation at 72 h in OE21 and KYSE140 cells, but combination of trametinib and erlotinib or afatinib treatment significantly affected cellular viability (Fig. 3c and Supplementary Fig. 9a,c). More formal synergy testing of trametinib and erlotinib in the OE21 cell line revealed a synergistic effect in the OE21 cell line but not KYSE140 (Supplementary Fig. 9b,d). Further investigation showed that 48 h of treatment with erlotinib, afatinib, trametinib or the combinations thereof increased G0/G1 arrest in OE21 and KYSE140 with the magnitude of cell cycle arrest greater with combination treatment (Fig. 3d). Furthermore, combination therapy led to greater annexin V-positive cells with 72 h of treatment, suggesting that MEK inhibition augmented apoptosis induced by EGFR inhibitors (Fig. 3e). Beyond combinations where both drugs are co-administered simultaneously, we also evaluated the timing of combination therapy. EGFR inhibitor erlotinib was dosed first, followed 24 h later by the addition of trametinib in combination with the EGFR agent. We also tested the reverse sequence with the trametinib administration 24 h before combination with erlotinib. These results demonstrated similar results with co-administration or the administration of EGFR blockade before MEK therapy. However, dosing of trametinib before EGFR inhibition appeared to modestly blunt the effect of the EGFR inhibition (Fig. 3c–e).

Based upon these results showing the potential joint efficacy of MEK and EGFR blockade and the role of ERK reactivation in response to EGFR inhibition in ESCC, we next tested whether

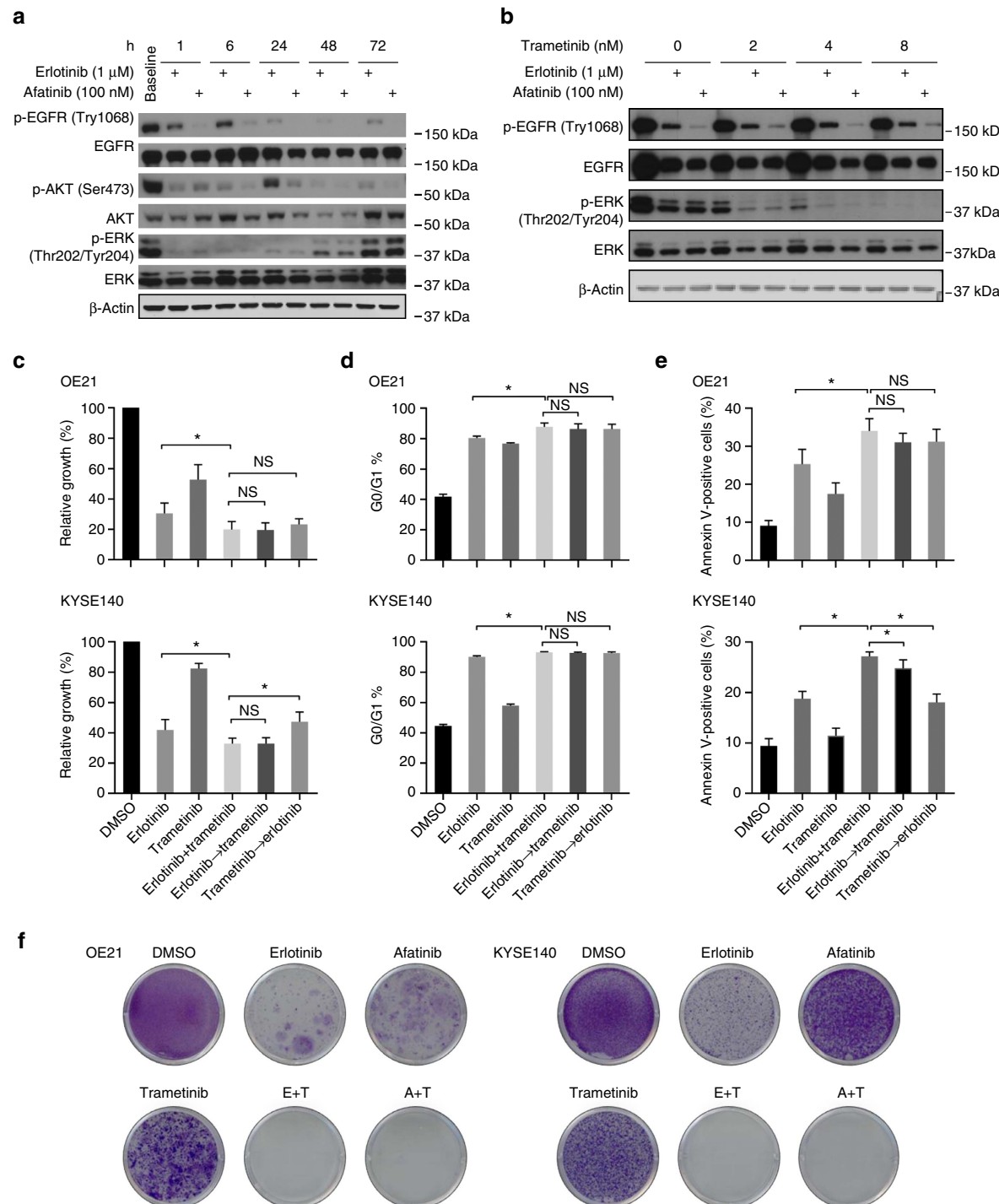

**Figure 3 | ERK reactivation following EGFR tyrosine kinase inhibitor (TKI) treatment facilitates resistance to EGFR inhibition.** (**a**) Immunoblots demonstrating the biochemical effects of distinct EGFR kinase inhibitors in OE21 cell line to erlotinib and afatinib at multiple time points after onset of therapy. (**b**) Immunoblots showing biochemical responses of OE21 cells to erlotinib alone or in combination with multiple doses of the MEK inhibitor trametinib. For erlotinib and trametinib combination, cells were treated with erlotinib first and were allowed to grow for 72 h. Then, erlotinib was added to the culture 6 h before protein harvest. (**c**) Plots depict the growth of OE21 and KYSE140 treated *in vitro* with either 1 μM erlotinib or 100 nM trametinib alone or in combination. All data are expressed as the percentage of growth relative to that of vehicle-treated control cells. (**d**) Plots depict the percentage of cells in G0/G1 following treatment with dimethylsulfoxide (DMSO), 1 μM erlotinib, 100 nM trametinib or a combination thereof for 48 h, with cell cycle status then assessed by flow cytometry. (**e**) Plots representing the induction of apoptosis, as measured by flow cytometry, in cells after 72 h of *in vitro* treatment with vehicle, 1 μM erlotinib and 100 nM trametinib or combination. For sequential strategy (**c**–**e**), cells were treated with one drug first and were allowed to grow for 24 h. Then, the second drug was added to the culture for another 24 h (for cell cycle analysis) and 48 h (for growth curve and apoptosis analysis). (**f**) Images show representative results of focus formation assays where cells were grown in culture in 6-well plates and treated with DMSO, erlotinib 1 μM, afatinib 100 nM and trametinib 10 nM weekly, and then fixed and stained with crystal violet solution after 4 weeks of treatment. All experiments were performed in triplicate for each condition and repeated at least twice. All error bars represent s.d., $n \geq 3$. Student's *t*-test was used for statistical analysis (*$P < 0.05$).

addition of MEK inhibition could prevent or delay the emergence of acquired resistance in OE21 and KYSE140 cells when treated with EGFR inhibitors. In both cellular models, monotherapy with erlotinib or afatinib led to the emergence of resistant colonies within 2–3 weeks. The emergence of such resistant clones was greatly inhibited or delayed by co-administration of 10 nM trametinib (Fig. 3f). No clones emerged in KYSE140 cells after as long as 12 weeks, whereas resistant clones of OE21 emerged after 6–8 weeks, demonstrating the potential value of co-administration of these agents as an up-front approach for ESCC therapy. When we evaluated resistant OE21 clones following erlotinib + trametinib or afatinib + trametinib treatment, we also found upregulation of EMT markers in those combined resistant cells (Supplementary Fig. 10).

We also evaluated the combination of MEK and EGFR blockade in our original cell lines that were insensitive to EGFR therapy, TE8, KYSE30 and KYSE520. Trametinib was able to augment erlotinib/afatinib sensitivity and successfully block the persistent phosphorylation of ERK in the KYSE30 cells that harbour the combination of *EGFR* amplification and *HRAS* mutation (Supplementary Fig. 11a,b). We also tested the impact of the addition of trametinib to erlotinib therapy in the TE8 and KYSE520 cell line and found only modest growth inhibition effect, consistent with the potential role of EMT as a mediator of resistance in these lines (Supplementary Fig. 11c,d). Broadly, these data point to the possible utility of dual EGFR/mitogen-activated protein kinase (MAPK) therapy in EGFR-positive ESCC but suggest that such combinations will not be uniformly efficacious, including after the emergence of EMT.

**CDK4/6 and EGFR inhibition block the emergence of resistance.** Beyond the potential of MAPK therapy to augment EGFR-directed therapy, we next aimed to explore additional agents that might augment EGFR blockade. Based upon genomic data from ESCC that have supported amplification of *CCND1* as a prominent feature in these cancers, we evaluated the co-occurrence of alterations in this pathway in genomic data from TCGA. In the genomes of 15 *EGFR*-amplified tumours, we queried the presence of alterations at *CCND1* and related cell cycle regulators, identifying co-occurring oncogenic alterations localizing of *CCND1*, *CDKN2A*, *CDK6* and *RB1* (Fig. 4a) in addition to *EGFR*. Among these, the most notable secondary alteration was the focal amplification at chromosome 11q13, at the locus of *CCND1*, in 10 samples (66.7%). We also observed deletion or truncating mutation of *CDKN2A* in 9 samples (60%), *RB1* deletion in 1 sample (6.7%) and *CDK6* amplification in 5 samples (33.3%). These data suggest joint EGFR amplification and cell cycle dysregulation are prominently co-occurring features in these tumours. Re-review of our models showed that the OE21 cell line harbours co-occurring amplification of *CCND1* and *CDKN2A* deletion, whereas the KYSE140 cell line harbours *CDKN2A* deletion in addition to *EGFR* amplification (Supplementary Table 2). These data raised hypotheses regarding possible augmentation of EGFR blockade with cyclin-dependent kinase 4/6 (CDK4/6) inhibition, analogous to data in breast cancer where such agents act synergistically with PI3K blockade[30].

We tested this hypothesis using the Food and Drug Administration (FDA)-approved CDK4/6 inhibitor palbociclib, first showing the ability of this agent to effectively inhibit CDK4/6 activity as determined by inhibition of Rb phosphorylation (Supplementary Fig. 12). In OE21, palbociclib monotherapy induced G0/G1 arrest, but showed more modest effects upon growth inhibition or apoptosis (Fig. 4b–d). Similarly, combined EGFR and CDK4/6 inhibition did not alter the effects of phosphorylation of downstream signalling mediators AKT, nor

did it block p-ERK rebound (Supplementary Fig. 12). Modest effects upon cell proliferation, cell cycle and apoptosis were shown with combination therapy (Fig. 5b–d and Supplementary Fig. 13a,c). KYSE140 showed a modest synergistic growth inhibition from the addition of palbociclib to erlotinib, although not as substantial as that observed in the OE21 model (Supplementary Fig. 13b,d). Similar to EGFR and MEK blockade, palbociclib given simultaneously with erlotinib, or after erlotinib, showed better growth inhibition than giving palbociclib before erlotinib (Fig. 4c,d). These data again caution against giving the secondary inhibitor, in this case targeting CDK4/6, before dosing with EGFR-directed therapy. Although the initial effects of combined EGFR and CDK4/6 inhibition in short-term viability assays were of only modest effects, we evaluated the effect of palbociclib upon the emergence of resistance to EGFR-directed monotherapy, and found that erlotinib + palbociclib showed the ability of blocking the onset of resistance *in vitro*. We could not generate any resistant clones to erlotinib combined with palbociclib in both OE21 and KYSE140 cells after 12 weeks of treatment. These studies demonstrated clear blockade of resistance with joint administration of the two drugs, suggesting that although this combination may not have as robust an impact upon initial response, it may prove an effective target to block resistance (Fig. 4e).

**CDK4/6/MEK inhibition improves erlotinib *in vivo* response.** To further evaluate EGFR inhibition alone or in combination with either MEK or CDK4/6 blockade, we next evaluated these therapies *in vivo* in nude mice harbouring xenografts of OE21 grown in their flanks. After tumours were established (having grown to ∼100–150 mm$^3$), mice were initiated on treatment with vehicle, erlotinib, trametinib, palbociclib or the combination of erlotinib with the MEK or CDK4/6 inhibitor. During the course of the 4-week treatment, single-agent erlotinib or trametinib delayed tumour growth, but progression still occurred. Palbociclib monotherapy interestingly allowed initial growth of the tumour but this was followed by subsequent stabilization of tumour volume in the following 3 weeks. Although the addition of trametinib to erlotinib slowed tumour outgrowth, only palbociclib/erlotinib combination showed consistent reduction in tumour volume (Fig. 5). Beyond the better apparent response to the EGFR and CDK4/6 combination, we also observed less apparent toxicity with this combination compared with the MEK doublet. Although weight loss was similar in the two groups, the MEK/EGFR combination led to skin changes (Supplementary Fig. 14) not seen with the palbociclib doublet. Immunohistochemistry analysis of OE21 xenografts demonstrated a modest decrease of Ki67 and increase of caspase 3 expression with erlotinib or trametinib single agent. Palbociclib monotherapy showed modestly decreased ki67 expression but no strong induction of caspase 3. The erlotinib/trametinib combination did enhance caspase 3 expression, signifying increased apoptosis, and modestly decreased ki67 repression. The erlotinib/palbociclib combination decreased tumour proliferation as evidenced by reduced Ki67 staining and increased caspase 3 expression (Supplementary Fig. 15). We tried to grade immunohistochemistry for E-cadherin, N-cadherin and vimentin, but could not find substantial difference between the erlotinib and vehicle groups (Supplementary Fig. 16). The discrepancy between *in vitro* and *in vivo* study may be because we use consistent dose of erlotinib during the *in vivo* experiment, whereas we did increase the dose gradually when we culture the resistant clones *in vitro*.

We also tested the erlotinib + trametinib and erlotinib + palbociclib combination in nonobese diabetic/severe combined immunodeficiency (NOD/SCID) mice bearing KYSE140 xenografts. These studies also showed that despite the continued

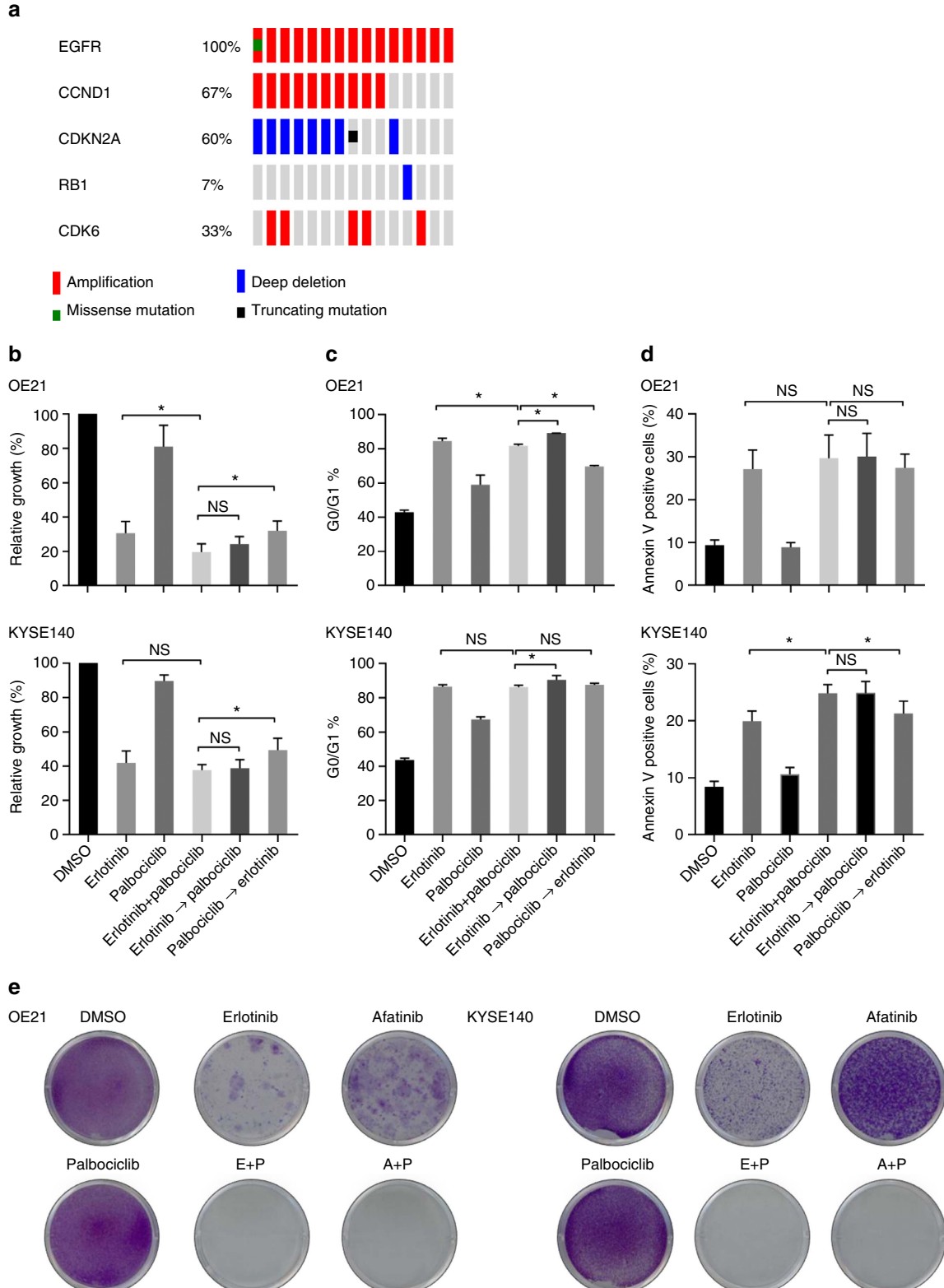

**Figure 4 | Blockade of CDK4/6 with EGFR prevents the emergence of resistance to EGFR inhibition.** (**a**) Integrated view of genomic aberrations of genes encoding cell cycle regulatory proteins in The Cancer Genome Atlas (TCGA) data in EGFR-amplified tumours from TCGA. Each column denotes an individual tumour, and each row displays a gene. Mutations are colour coded by the type of mutation, and amplifications are depicted as red outlines. (**b**) *In vitro* growth inhibition of OE21 and KYSE140 cell lines following treatment with erlotinib 1 μM, palbociclib 1 μM or a combination thereof for 72 h or sequential treatment. The flow cytometry (**c**,**d**) as well as crystal violet (**e**) assays for these combination treatment studies are shown as in Fig. 3. All experiments were performed in triplicate for each condition and repeated at least twice. Student's *t*-test was used for statistical analysis (*$P < 0.05$). All error bars represent s.d., $n \geq 3$.

**a**

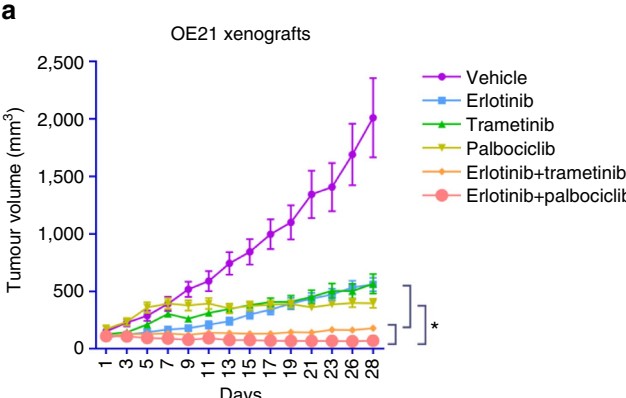

**b**

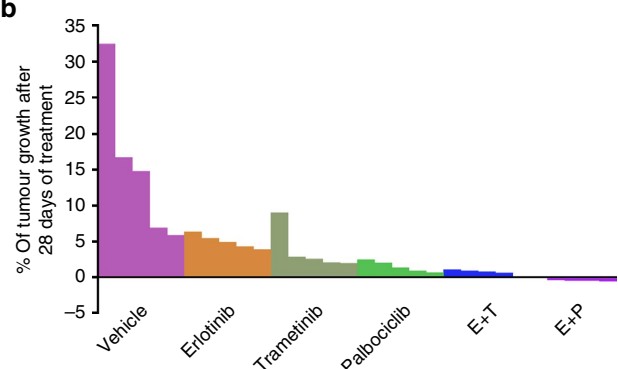

**Figure 5 | CDK4/6 or MEK inhibition improves erlotinib response in ESCC xenografts.** (**a**) Curves plot the growth of OE21 xenograft tumours were treated with vehicle control, erlotinib, trametinib, palbociclib, erlotinib + trametinib or erlotinib + palbociclib (mean ± s.e.m., 5 mice per condition). (**b**) Waterfall plot showing the percentage change in volume for the individual tumours in each arm at treatment day 28. Tumour volumes were normalized individually to their volumes at treatment day 1. Student's t-test was used for statistical analysis (*$P < 0.05$).

growth of tumours with erlotinib, trametinib or palbociclib, treatment with the combination of these agents is effective *in vivo* (Supplementary Fig. 17). These *in vivo* data add to the support for testing of the EGFR and CDK4/6 combination in patients with *EGFR*-positive ESCC as well as for the potential efficacy of the EGFR/MEK doublets.

## Discussion

Despite the presence of multiple FDA-approved inhibitors to EGFR and the documented presence of *EGFR* amplification as a common feature of ESCC, treatment of systemic disease remains reliant upon cytotoxic therapy. Our data demonstrate the importance of revisiting the potential to target EGFR in the care of these patients. Initial studies investigating this target suffered from both the failure to utilize biomarkers to select patients with somatic alterations of *EGFR* and from the reliance upon monotherapy against EGFR as a targeting strategy. Here, we demonstrate that multiple *EGFR*-amplified ESCC cellular models show clear sensitivity to EGFR blockade with small-molecule inhibitors. These data speak of the potential to further exploit this target in patients.

However, although there is clear potential for targeting EGFR, these data also speak of significant limitations. As with all targeted therapies, acquired resistance to targeted therapeutic drugs remains a major obstacle in cancer medicine[31]. Notably, our results with acquired resistance found that we were readily

able to generate resistance to EGFR blockade and also that this resistance could occur with reactivation of downstream pathways and a transition to an EMT-like state lacking clear dependence upon EGFR. EMT transitions have been observed in the setting of acquired resistance to EGFR-directed therapy, exclusive to other identified resistance mechanisms in cell line models and patients[24,27,32–34]. The molecular mechanisms connecting the resistance of the cancer cells to the mesenchymal phenotype remain unknown, but such transitions have been shown to abrogate sensitivity to blockade of oncogene drivers[18,21,24,35]. Although investigation of mechanisms of combating acquired resistance was not the goal of this study, we followed our observations of upregulation of AXL in the setting of EMT and resistance. Studies suggest that blockade of AXL may be able to augment EGFR blockade in the setting of tumours with EMT-induced resistance or, potentially, tumours with pre-existing mesenchymal phenotypes[21,24,35]. However, we found only AXL inhibitor R428 showed some single-agent activity, but was not substantial augmentation of EGFR inhibitor. Genetic inhibitory effect of AXL did not enhance the antiproliferative role in ESCC as pharmacologic effect. The discrepancy for pharmacological and genetic inhibitory effect require further study in order to evaluate the possibility of AXL blockade in this disease. More broadly, these data speak of the clear challenges associated with targeted therapies in tumours with a mesenchymal state.

Although more studies are ultimately needed in order to fully investigate the EMT phenotypes and optimal means to target such resistant tumours, another key conclusion from these resistance studies is that it will be optimal to develop up-front treatments that block the emergence of resistance rather than to treat resistance after its emergence. Indeed, among EGFR-mutant lung cancers with acquired resistance to their initial EGFR inhibitor, there is growing evidence that all resistant cells may not share the same, but exhibit different resistance mechanisms, likely reflecting both intratumoural and intertumoural heterogeneity, as well as dynamic changes in the relative populations of resistant clones over time[14,36,37]. This problem adds to the potential benefit of blocking resistance with up-front combination approaches. In this setting, we observe reactivation of ERK1/2 as an adaptive change after EGFR inhibition in EGFR-sensitive ESCC lines, similar to data seen in the study of EGFR inhibition of non-small-cell lung cancer[29]. In colorectal cancer, acquired drug resistance to EGFR antibody cetuximab is also being observed to converge upon ERK reactivation that, in turn, could be rationally targeted by further lines of therapy[38]. Most importantly, these data strongly suggest the potential therapeutic benefit to dual blockade of the MAPK pathway with EGFR. Following upon that idea, we showed that MEK/EGFR blockade can delay the emergence of resistance induced by EGFR inhibitors.

These data on MEK/EGFR combinations provide rationale to evaluate such combinations clinically in ESCC. However, there are limitations to this type of combination, including the presence of overlapping toxicities from these classes of inhibitors. Therefore, we searched for other possible targets whose use may be able to enhance the efficacy of EGFR blockade. Looking at genomic data from EGFR-positive ESCC, we noted striking co-occurrence of somatic alterations of cell cycle regulators. As cell cycle regulatory factors altered in ESCC are, in a simplified model of cellular physiology, downstream of the MAPK pathway, these data suggest that targeting of the cell cycle could have similar effects in combination with EGFR blockade as seen with MEK inhibitors. Although our *in vitro* data suggest that the initial effects of the FDA-approved inhibitor palbociclib in combination with EGFR blockade are not as strong as seen with the MEK/EGFR combination, the addition of CDK4/6 inhibitor can

block the emergence of resistance to EGFR inhibition *in vitro* and appear to have more favourable effects *in vivo* compared with the MEK combination. As CDK4/6 and EGFR inhibitor combination may have benefits in terms of toxicity and tolerability, these data provide another candidate therapeutic approach to pursue in these patients.

The realization that relapsed tumours are highly molecularly heterogeneous poses a formidable therapeutic challenge, as it would seem quite difficult to overcome the multiple resistance mutations that arise in individual patients[38]. It is tempting to speculate that the best strategy to produce a lasting effect on cancers is to treat with combinations up-front to prevent potentially resistant clones from emerging. In a phase II clinical trial, combinations of targeted agents conferred advantages over sequential treatments in melanoma patients treated concomitantly with anti-BRAF and anti-MEK drugs[25]. In addition, initial co-targeting of EGFR and MEK has been shown experimentally to impede the development of acquired resistance in EGFR-mutant lung cancer[15].

Here, we propose that EGFR-amplified ESCC may develop early adaptive response to EGFR inhibition by activating MAPK pathway in a rapid manner. Co-targeting EGFR and MEK can prevent or delay the emergence of a potentially broad variety of drug resistance mechanisms. Notably, in the OE21 model, the ultimate resistance followed emergence of EMT, leading to a state where MAPK blockade no longer greatly potentiated erlotinib therapy. Although MAPK reactivation was not the ultimate aetiology of acquired resistance, its blockade could thwart the emergence of cells with the mesenchymal-resistant phenotype. Reactivation of MAPK signalling is thus likely independent of the mechanisms driving acquired resistance[31,38–40] and thus likely enables cancer cells to survive and subsequently activate other bypass routes for survival and proliferation. These new data provide a rationale for overcoming resistance to EGFR inhibitor using MEK inhibitors, many of which have already reached the clinic[41].

We showed that amplifications of cell cycle-related genes such as *CCND1* and *CDK4/6* and deletion of *CDKN2A* gene co-exist in considerable proportions of EGFR-amplified ESCC. Our study advances the strategy of joint delivery of EGFR and cell cycle inhibitors, and highlights the enhanced efficacy of the erlotinib with palbociclib combination relative to erlotinib monotherapy as a novel mechanism of enhancing efficacy of anti-EGFR therapy. Although the addition of palbociclib did not appear to directly augment the inhibition of the intermediate cell signalling pathways downstream of EGFR as seen with the addition of MEK therapy to erlotinib, it was able to similarly block the emergence of resistance to EGFR blockade. Concomitant inhibition of CDK4/6 with PI3K inhibitors has been shown to enhance sensitivity in preclinical breast cancer models[30], suggesting that this capacity is not EGFR inhibitor specific. Our findings highlight cell cycle regulation as a critical point of convergence for multiple genetic alterations, and provide a significant therapeutic rationale for clinical evaluation of concomitant EGFR and CDK4/6 blockade in patients with EGFR-amplified ESCC. Given the frequency that these targets are altered and the availability of inhibitors, the potential to speed development of new therapies for ESCC is substantial.

In conclusion, genomic amplifications of the gene encoding *EGFR* in ESCC has a clear potential to serve as biomarker to guide the use of targeted inhibitors. Through testing of therapeutics in genomically defined model systems, we have been able to identify candidate rational combinations of targeted agents for enhanced efficacy of EGFR in ESCC. We hope that these data will motivate clinical studies of EGFR inhibitors in combination with other complementary inhibitors in genomically defined ESCC patients.

## Methods

**Genomic characterization of human samples and cell lines.** To interrogate the significantly co-occurring oncogenic copy-number alterations in EGFR-amplified ESCC, we queried single-nucleotide polymorphism microarray data publically available from The Cancer Genome Atlas (TCGA) that had been processed and had focal gene amplifications identified as per TCGA protocols[42] (http://cancergenome.nih.gov/). Informed consent was obtained from all human participants who contributed samples for the TCGA effort.

TE8 and TE10 cell lines were obtained from the University of Pennsylvania. OE21, KYSE30, KYSE70, KYSE140, KYSE180, KYSE450 and KYSE520 were obtained from The Broad Institute. All cell line genomic characterization was obtained from the Cancer Cell Line Encyclopedia project (http://www.broadinstitute.org/ccle/home). KYSE450 was maintained in RPMI-1640/F12 (1:1) supplemented with 10% fetal bovine serum (FBS) and 1% penicillin/streptomycin/L-glutamine. All other cells and drug-resistant OE21ER, OE21AR were cultured with RPMI-1640 supplemented with 10% FBS and penicillin/streptomycin/L-glutamine. All cells were kept in a humidified incubator at 5% $CO_2$. Small-molecule inhibitors such as erlotinib (S1023), afatinib (S1011), trametinib (S2673), palbociclib (S1116), GDC0941 (S1065) and R428 (S2841) were purchased from Selleck Chemicals (Houston, TX, USA). All drugs were prepared as 5–10 mM stock solutions in dimethylsulfoxide (DMSO) and stored at $-20\,°C$.

Resistant cell lines were generated using parental OE21 cells by culturing with stepwise escalation of concentrations of erlotinib (500 nM) or afatinib (50 nM) starting in week 5, until a concentration of erlotinib (5 µM) or afatinib (500 nM) was reached at the end of a 30-week period. Single-cell cloning from OE21ER was performed using flow cytometer sorting and confirmed to be drug resistant. Identity of the resistant cells was confirmed via STR genotyping (IDEXX Bioresearch).

**Cell proliferation and viability assays.** Cell viability was measured with the CellTiter-Glo luminescent Cell Viability assay (Promega, Madison, WI, USA) according to the manufacturer's instructions. Briefly, cells were plated at a desired density (1,500–3,000 cells per well) onto flat-bottomed 96-well plate. After 24 h, cells were treated with either vehicle (DMSO) or variable doses of small-molecule inhibitors and then allowed them to grow for 3 days. Then, the relative amount of ATP was measured using a luminometer. All experiments had three technical replicates and three biological replicates. Data were expressed as percentages of the survival of control (DMSO treated) cells, calculated from the absorbance corrected for background. Synergy testing was performed using the Chou-Talalay method. Briefly, 5 different concentrations of each drug ($0.0625\times$, $0.25\times$, $1\times$, $4\times$ and $16\times$ $IC_{50}$) were given either alone or in combination, maintaining a constant ratio. Cell viability was measured using CellTiter-Glo, and the results were analysed using COMPUSYN.

**Antibodies and western blotting.** Cells were plated at $2–3\times10^5$ cells per well in 6 cm plates for assessment of EGFR and downstream signalling pathway protein expression. Cells were washed twice with ice-cold phosphate-buffered saline (PBS) and lysed with RIPA lysis buffer (50 mM Tris-HCl, pH 7.5, 150 mM NaCl, 0.1% SDS, 1% NP-4, 0.5% sodium deoxycholate; Boston Bioproducts) supplemented by protease inhibition cocktail (Roche) and phosphotase inhibitor cocktails (BD). Lysates were separated on 4–12% Tris-Glycine SDS–polyacrylamide gel and were transferred to PVDF membranes (Invitrogen). The membranes were blocked with 5% skim milk (Bio-Rad) dissolved in TBST buffer (50 mM Tris-HCl, 150 mM NaCl, Tween-20). The membranes were incubated with primary antibodies overnight at 4 °C. The following antibodies were used for western blotting (all from Cell Signaling Technologies, Beverly, MA, USA, except where indicated): anti-phospho EGFR Tyr 1068 (3777, 1:2,000), anti-EGFR (4267, 1:6,000), anti-phospho AKT Ser-473 (4060, 1:500), total AKT (9272, 1:1,000), anti-phospho ERK1/2 Thr 202/204 (4370, 1:500), anti-ERK1/2 (4695, 1:1,000), anti-vimentin (5741, 1:1,000), anti-SOX2 (14962, 1:1,000), anti-AXL (8661, 1:1,000), anti-phospho S6RP Ser235/236 (2211, 1:1,000), anti-S6 (2217, 1:2,000), anti-phospho RB Ser807/811 (9308, 1:500) and anti-Rb (9309, 1:1,000). Anti-N-cadherin (BDB610920, 1:1,000) and anti-E-cadherin (BD 610181, 1:1,000) were obtained from BD Biosciences. β-Actin (1:20,000) was obtained from Sigma Aldrich (A5441, St Louis, MO, USA). Horseradish peroxidase-conjugated secondary antibodies (anti-rabbit sc-2004, anti-mouse sc-2055, Santa Cruz, Santa Cruz, CA, USA) and SuperSignal West Pico Chemiluminescent Substrate (34080, Life Technologies) were used to detect signals. Western blot analyses were performed at least twice, starting with independent cell lysates. Uncropped blots are shown in Supplementary Fig. 18.

**AXL knockdown.** Transfection of short interfering RNA (siRNA) against AXL was performed using siRNA for the AXL gene (ON-TARGETplus Human AXL siRNA, l-003104-00-0010). Cells were plated in a 6-well plate and transfected at a density of $2\times10^5$ per well with 20 nM siRNA following complexation with 6 µl

Lipofectamine RNAiMAX (Invitrogen, 13778150) according to the manufacturer's instructions.

**Apoptosis.** Cells were exposed to inhibitors or DMSO for 72 h and harvested. FITC annexin V was used for apoptosis assessment (FITC annexin V Apoptosis Detection Kit I, Becton Dickinson, Cat. No. 556547). Briefly, after washing twice with cold PBS, $1 \times 10^5$ cells were resuspended in $1 \times$ binding buffer. Cells were then stained with FITC annexin V and propidium iodide, analysed by fluorescence-activated cell-sorting on an LSR II flow cytometer (BD Biosciences), and data were assessed with FlowJO software (TreeStar).

**Cell cycle analysis.** For cell cycle analyses, cells were plated and treated the following day with the indicated agents. After 48 h of treatment, cells were harvested and stained with Cycle TEST PLUS DNA Reagent Kit (Becton Dickinson, Cat. No. 340242) according to the manufacturer's recommendations. Briefly, after washing with ice-cold PBS, cells were harvested and washed with Buffer solution for 3 times. $5 \times 10^5$ cells were needed and gently mixed with trypsin buffer (Solution A), trypsin inhibitor and RNase buffer (Solution B), propidium iodide stain solution (solution C) step by step. The fluorescence-activated cell-sorting analysis was performed on the LSR II flow cytometer mentioned above and data were analysed with ModFIT LT software.

**Crystal violet.** To measure the emergence of acquired resistance, cell lines were plated in triplicate 6-well plates ($2 \times 10^5$ cells per well) and were treated with different inhibitors or DMSO after 24 h of seeding, and then treatments with fresh media were exchanged every 5–7 days thereafter. Cells were stained with Crystal Violet (V5265, Sigma, St Louis, MO, USA) at 2 and 4 weeks or when cells grew confluent. After washing twice with PBS, cells were fixed with 1% paraformaldehyde and incubated for 15 min at room temperature. Then, cells were washed two more times with PBS and stained with 1% crystal violet for 15 min at room temperature.

**Mouse cohorts and treatment.** All care and treatment of experimental animals were conducted under a protocol approved by the Harvard Medical School/Dana-Farber Cancer Institute (DFCI) institutional animal care and use committee guidelines. All mice were housed in a pathogen-free environment at DFCI animal facility and handled in strict accordance with Good Animal Practice as defined by the Office of Laboratory Animal Welfare.

Nu/Nu mice and NOD/SCID mice were purchased from Jackson Laboratory (Bar Harbor, ME, USA). OE21 and KYSE140 cell line were detected as pathogen free and cultured in RPMI-1640 with 10% FBS. The cells were washed with serum-free medium and resuspended in serum-free medium mixed with an equal amount of Matrigel (BD Biosciences). Mice were injected with 2 million cells (OE21) and 10 million cells (KYSE140) per injection with two distinct injections in the flank of each mouse. The mice were randomly grouped and treatment was started when the tumours size reached 100 to 150 mm³. Each cohort included at least 5 mice (6–10 tumours). Tumour sizes were monitored twice weekly and volumes were calculated with the formula: $(mm^3) = length \times width \times width \times 0.5$.

Erlotinib was dissolved in 0.5% methyl cellulose with 0.4% Tween 80; trametinib were dissolved in 0.5% hydroxypropyl methylcellulose with 0.4% Tween 80; palbociclib was dissolved in 17% (2-Hydroxypropyl)-β-cyclodextrin. Erlotinib was dosed as 50 mg kg$^{-1}$ daily, trametinib was given as 2 mg kg$^{-1}$ daily and palbociclib was dosed as 100 mg kg$^{-1}$ daily. All inhibitors were given via gavage.

**Statistical analysis.** Statistical analysis was performed with SPSS and GraphPad Prism software. IC$_{50}$ values were obtained using GraphPad Prism software (La Jolla, CA, USA). Comparisons between experimental arms were performed by Student's t tests, with P values of $<0.05$ considered significant. For all in vitro functional experiments, data are expressed with mean ± s.d. For in vivo experiments, statistical comparison among groups are carried out with one-way analysis of variance Kruskal–Wallis test, and are presented as the mean ± s.e.m.

**Data availability.** All data generated or analysed during this study are included in this published article (and its Supplementary Information files), and all relevant data are available from the authors. No genomic data sets were generated during the current study.

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

## Acknowledgements

We thank N. Sethi and other members of the Bass laboratory, and X. Zhang of the Meyerson laboratory for discussions. A.J.B. was supported by the NIH Grants R01 CA187119 and R01 CA196932. A.K.R., J.A.D., K.-K.W. and A.J.B. were supported by the NIH Grant P01 CA098101 and its Molecular Pathology/Imaging and Molecular Biology/ Gene Expression Core Facilities. A.K.R. was also supported by the NIH Grants R01 DK060694 and R01 DK056645, American Cancer Society RP-10-033-01-CCE and P30 DK050306.

## Author contributions

A.J.B. and J.Z. designed the study. J.Z., Z.W., G.W., E.P., A.N., M.S., H.Z, T.C., J.B.L. and X.X. performed experiments. F.S.-V. provided computational assistance. A.K.R, K.-K.W. and J.A.D. provided expert consultation. A.J.B. and J.Z. wrote the manuscript. A.J.B. supervised the study.

## Additional information

**Competing financial interests:** The authors declare no competing financial interests.

**Reprints and permission** information is available online at http://npg.nature.com/ reprintsandpermissions/

**Publisher's note**: 

