## [Peer Review File · Nature Communications]

Reviewers' comments:

Reviewer #1 (Remarks to the Author):

In the manuscript by Zhou et al, the authors demonstrate that cell cycle regulator CDK4/6 or MAPK Blockade are able to enhance efficacy of EGFR inhibition in esophageal squamous cell carcinoma (ESCC). Overall, the paper presents a novel story. Which might be interesting to readers in the field of ESCC. However, I have listed my concerns below.

1. In the figure 1b, the expression levels of EGFR and p-EGFR were very low in K140 cells compared with the other cell lines, author should add a normal cell group.
2. In the figure 1c, figure 2b, what did the concentration of "-4" and "-2" represent in the X-axis?
3. There were many inconsistent logograms, such as "p EGFR" and "p-EGFR", "ERK" and "ERK1/2" and so on.
4. Which EGFR phosphorylation site was represented in the result? Because different EGFR phosphorylation sites display different roles.
5. In the figure 1d, cells were harvested at three time points (1h, 6h, and 24h) after treatment with 1uM Erlotinib or 100nM Afatinib in OE21 and K140 cell lines, why cells were harvested at 6h in OE21, K30, K520 and TE8 cell lines in sup. figure 2?
6. In the second paragraph of fifth page, the author described "EGFR phosphorylation was partially blocked by 1uM erlotinib and completely blocked by 100nM afatinib treatment in all cell lines", but according to the results in the figure 1d, EGFR phosphorylation was also partially blocked by 100nM afatinib treatment in OE21 cells, and EGFR phosphorylation was completely blocked by 1uM erlotinib at 6h in K140 cell. Furthermore, the author described "However, downstream signaling persisted in the resistant TE8, KYSE30 and KYSE520 cell lines (Supplementary fig. 2)", but EGFR downstream signaling (p-AKT and p-ERK) were also partially inhibited in K30 cells in sup. figure 2. The author should explain the reason.
7. The second sentence lacked references in the third page second paragraph.
8. In the figure 1e-f, which concentrations were chosen? And the figure 1e-f, sup. figure 7 also lacked significantly statistical analysis.
9. The western blot results of EGFR, p-EGFR and actin in figure 1b were duplicated to sup. figure 1, the author should change them.
10. The picture lacked scale in figure 3a
11. OE21ER cells had a morphology change compared with OE21 parental cells, but the author did not show if the OE21 AR cells also had a morphology change compared the control cells before displaying the figure 3b results.
12. In the second paragraph of seventh page, the author described "We noticed that the resistant cell population was heterogeneous and showed different morphology and growth rate", but I did not see any data.
13. The figure 3d lacked MK2206 and rapamycin results; figure 3d,e lacked DMSO group,
14. In the figure 3d, what the drug concentrations were chosen in trametinib and GDC0941 treatment?
15. The author described "These inhibitors did not have substantial effect in OE21ER cells despite adequate biochemical target engagement (fig. 3 d, e, Supplementary Fig. 4)", but as shown in figure 3e, we found Trametinib can significantly inhibit OE21ER cells growth, while Rapamycin also had a substantial effect in OE21ER cell growth in sup. figure 4. The author should explain the reason.
16. In figure 4a, the results lacked DMSO group; the author described "ERK1/2 reactivation was observed within just 48 hours following continuous exposure to treatment with erlotinib or afatinib, despite a lack of return of the phosphorylation of AKT or EGFR (Fig. 4a)." In fact, as shown in figure 4a, phosphorylation of AKT reactivation was also observed within just 48 hours following continuous exposure to treatment with erlotinib or afatinib. The author should explain the reason.
17. Sup, figure 6 had disordered composition; ERK1/2 reactivation was not observed within 48 hours following continuous exposure to treatment with erlotinib or afatinib in K140 cells, while author described "Other EGFR inhibitor sensitive-lines showed similar ERK reactivation (Supplementary Fig. 6)." The author should explain the reason.

18. Sup. figure 7 did not mark the columns to represent different inhibitors.
19. In figure 4c,d,e, why occurred different treated times for the same inhibitors. How many concentrations were chose for each inhibitors?
20. Sup, figure 8 had disordered composition; it did not mark the columns to represent different inhibitors, and also lacked significantly statistical analysis.
21. In sup. figure 9, I did not see the results about Rb phosphorylation;
22. I did not find the "figure 6" in the text.
23. The statistical methods described here were not integrated.
24. I urge the authors to invite some native English-speaking scientists to revise the manuscript.

Reviewer #2 (Remarks to the Author):

The authors describe progress in identify drug combinations for use in ESCC tumor cell lines bearing EGFR amplifications, a common driver of this disease. Although some such tumors respond to EGFR inhibitor therapy initially, resistance develops, and others are de novo resistant. The authors show that downstream signaling can be reactivated, as in numerous other tumor types, and MEK inhibition, for example, can be helpful. However, once resistance develops, it often involves EMT, and such cells are poorly responsive even to combination therapies. The authors go on to show that EGFR amplified ESCC often contain RB pathway mutations, almost always involving loss of p16 or amplification of cyclin D or cdk6, consistent with numerous previous studies of OSCC. Interestingly, cdk4/6 inhibitors can somewhat augment EGFR inhibitors in cell culture with arrest and apoptosis endpoints, but have a more robust effect on preventing colony outgrowth when given in combination, and are very effective at preventing xenograft growth of one cell line. The take home message is that combinatorial therapy with cdk4/6 inhibition could prevent outgrowth of resistant cells, and thus the generation of highly intractable, EMT cells. Overall the study is interesting and supportive of the concept of a clinical trial for this combination. On the downside, few new concepts are uncovered and the efficacy of the combination is not surprising in light of other studies, including the authors' reference 25, interrogation PI3K + cdk4/6 inhibition in breast cancer. However, given the potential clinical implications of this study, it is nevertheless likely to be of significant impact. The following issues are important to address:

1. The response of OE21 and KYSE140 to palbociclib is quite different molecularly, but (apparently) not in terms of cell outgrowth. It is not clear if this reflects differences in response to palbociclib, because contrary to the text, no data for RB phosphorylation are presented in figure S9, so this cannot be assessed. This critical point needs to be addressed before a full evaluation is possible. On a more minor note, it is not clear if Figure 4f and 5e present the two cell lines as this in not labeled, but this is assumed to be the case.

2. It is a bit concerning that only one of the two cell lines is tested in vivo, given the dramatic response in vivo to dual inhibition that seems disproportionate to the proliferation and apoptosis response. Is a strong in vivo response also seen with KYSE140? Fundamentally, it is important to know that more that one cell line responds in this way, and ideally more than two, if molecular responses are different.

3. Pathologic analysis of tumors in figure 10 should be included to assess if combinations are augments apoptosis, arrest, both, etc.

minor point - RB pathway mutations are described in supplemental table 2, not 1, as called out in the text.

Reviewer #3 (Remarks to the Author):

In this study the authors addressed the mechanisms of resistance of esophageal squamous cell carcinoma (ESCC) cell lines to EGFR tyrosine kinase inhibitors (TKI). The authors demonstrate that epithelial mesenchymal transition (EMT) is involved in the acquired resistance to these drugs. They also show that the combination of EGFR TKIs with either MEK or CDK4/6 inhibitors can delay the occurrence of resistance and has a significant effect on the in vivo growth of EGFR TKI sensitive ESCC cells.

Overall, this is an interesting paper providing novel information on the possibility to develop target therapy in ESCC. However, the study also has several limitations and points to be clarified:

1) The authors found that EMT is involved in the acquired and intrinsic resistance to EGFR TKI (at least in some cell lines), but they do not follow up with these data. Rather than trying to identify the molecular mechanism involved in this phenomenon and potential targets for therapeutic intervention, the authors focused on the combination of EGFR TKI and MEK inhibitors to prevent the resistance. I did not find very logical way in which they decided to proceed, and the paper will be improved by additional characterization of resistant cells. In this regard, it will be better to generate additional resistant clones from different cell lines i.e. starting from a different genetic background.

2) The authors show in fig 4a a rebound of ERK1/2 activation following continuous exposure to erlotinib. This experiment needs to be better explained: did the authors change the medium during the experiment? Was erlotinib added during the experiment or cells were treated only once? These factors can make a significant difference on the results of this type of experiments.

3) The combination of trametinib and erlotinib does not seem much more effective as compared with erlotinib alone in fig 4c. This is also true for many other combination experiments showed in the paper. If the authors want to claim that a synergism occurs with these combinations, they need to perform isobolographic analysis of drug combinations.

4) The combinations of erlotinib with either trametinib or palbociclib seem to delay the occurrence of resistant clones. However, the authors should clarify whether resistant clones do develop at a certain time. In addition, it would be important to know whether the resistant clones undergo EMT as a major mechanism of resistance.

5) In the in vivo experiments, the combinations of erlotinib with either trametinib or palbociclib seem to be more effective as compared with a single drug. However, is the difference statistically significant?

6) The authors should analyze tumors from mice treated with single agent in order to assess whether EMT is induced also in vivo.

7) The manuscript should be checked for a number of minor errors affecting its quality. For example, in the text the author refer to supplementary table 1 for the genetic features of cell lines that are described in supplementary table 2. In addition, the author refer to Rb phosphorylation shown in Supplementary fig 9. The version that I received of this figure does not contain blots for Rb phosphorylation.

Response to Reviewers

We thank the reviewers for their encouraging and thoughtful comments and suggestions regarding our original submission. In response to these comments, we have made a number of modifications to our manuscript and performed additional suggested experiments. Below we detail these specific modifications with the specific comments from the reviewers (in blue text) followed by our response (in black). We hope the reviewers will find this manuscript improved following these changes and more suitable for publication in *Nature Communications*.

Reviewer#1 (ESCC)

In the manuscript by Zhou et al, the authors demonstrate that cell cycle regulator CDK4/6 or MAPK Blockade are able to enhance efficacy of EGFR inhibition in esophageal squamous cell carcinoma (ESCC). Overall, the paper presents a novel story. Which might be interesting to readers in the field of ESCC. However, I have listed my concerns below.

We thank the reviewer for these helpful comments and we have now added and edited important details to our experimental information. Specific concerns and comments are addressed below:

1. In the figure 1b, the expression levels of EGFR and p-EGFR were very low in K140 cells compared with the other cell lines, author should add a normal cell group.

We agree with this suggestion to add other different genetic background ESCC cell lines. In the revision version, we have now included TE10 and KYSE70 cells, which are EGFR non-amplified ESCC cell lines. The copy number and expression of p-EGFR and EGFR of each cancer cell line is shown in Figure 1b. These demonstrate that although p-EGFR and EGFR expression are variable in different EGFR-amplified cell lines, it is consistently higher than when compared with EGFR non-amplified ESCC lines.

2. In the figure 1c, figure 2b, what did the concentration of "-4" and "-2" represent in the X-axis?

The concentration -4 and -2 represent in the X-axis signify concentrations of 0.1nM and 10nM. We changed the axis in our figures.

3. There were many inconsistent logograms, such as "p EGFR" and "p-EGFR", "ERK" and "ERK1/2" and so on.

We now have made these abbreviations consistent through the text and figures.

4. Which EGFR phosphorylation site was represented in the result? Because different EGFR phosphorylation sites display different roles.

We use Tyr1068 phosphorylation site in all our experiments and have noted that in the methods.

5. In the figure 1d, cells were harvested at three time points (1h, 6h, and 24h) after treatment with 1uM Erlotinib or 100nM Afatinib in OE21 and K140 cell lines, why cells were harvested at 6h in OE21, K30, K520 and TE8 cell lines in sup. figure 2?

For OE21 and KYSE140, which is sensitive to EGFR inhibition, we performed a time course so we could investigate how signaling changed following EGFR blockade. In pilot studies (not shown) we investigated the time course of signaling in the EGFR inhibitor insensitive models and identified no difference. Therefore, we showed the 6hr time point as representative demonstration of drug effect.

6. In the second paragraph of fifth page, the author described "EGFR phosphorylation was partially blocked by 1uM erlotinib and completely blocked by 100nM afatinib treatment in all cell lines", but according to the results in the figure 1d, EGFR phosphorylation was also partially blocked by 100nM afatinib treatment in OE21 cells, and EGFR phosphorylation was completely blocked by 1uM erlotinib at 6h in K140 cell. Furthermore, the author described "However, downstream signaling persisted in the resistant TE8, KYSE30 and KYSE520 cell lines (Supplementary fig. 2)", but EGFR downstream signaling (p-AKT and p-ERK) were also partially inhibited in K30 cells in sup. figure 2. The author should explain the reason.

We agree with the reviewer and have tried to describe more accurately in our manuscript and figure legend: EGFR phosphorylation was modestly blocked by 1uM erlotinib and strongly blocked by 100nM afatinib treatment in all cell lines, and the phosphorylation of AKT and

ERK was clearly inhibited in the erlotinib/afatinib - sensitive lines OE21 and KYSE140. However, downstream signaling persisted or was only slightly reduced in the EGFR-inhibitor resistant TE8, KYSE30 and KYSE520 cell lines. These data imply that signaling molecules other than EGFR are critical for driving signaling to downstream pathways in these cancers although there is some degree of EGFR-driven activation (thus leading the modest effects upon p-AKT and p-ERK).

7. The second sentence lacked references in the third page second paragraph.

We thank the reviewer for noting this error. The appropriate references have been added into the manuscript.

8. In the figure 1e-f, which concentrations were chosen? And the figure 1e-f, sup. figure 7 also lacked significantly statistical analysis.

We have modified the manuscript to detail that we used erlotinib 1 μ M, or afatinib 100nM in the experiments represented in Figures 1e-f. Moreover, we have added the statistical testing demonstrating significant differences into the relevant figures. Drug dosing and the statistical tests are noted in the revised Figure Legends.

9. The western blot results of EGFR, p-EGFR and actin in figure 1b were duplicated to sup. figure 1, the author should change them.

We apologize for this oversight and have updated the western blot results of EGFR and p-EGFR in figure 1b (with the previously noted addition of the non EGFR-amplified ESCC cell line models).

10. The picture lacked scale in figure 3a

The scale has been added in all pictures of different cell lines.

11. OE21ER cells had a morphology change compared with OE21 parental cells, but the author did not show if the OE21 AR cells also had a morphology change compared the control cells before displaying the figure 3b results.

We have now included the figures of OE21AR cells in our figure 2c. These results demonstrate OE21AR cells showed similar morphology change as OE21ER cells.

12. In the second paragraph of seventh page, the author described "We noticed that the resistant cell population was heterogeneous and showed different morphology and growth rate", but I did not see any data.

The growth rate curve and morphology of different subclones have been added in our supplementary Figures 3b, c. These data demonstrate the heterogeneous characteristics between our subclones.

13. The figure 3d lacked MK2206 and rapamycin results; figure 3d,e lacked DMSO group,

All in vitro experiments were performed with DMSO control groups. We have made sure to include all of these necessary images of the controls in the modified Figures. Furthermore, we have now included the target engagement experiment with MK2206 and rapamycin inhibition in supplementary figure 5c.

14. In the figure 3d, what the drug concentrations were chosen in trametinib and GDC0941 treatment?

The concentrations of these agents have been included into the figures and figure legends. We show that these drugs were used at concentrations of 100nM for trametinib and 1uM for GDC-0941.

15. The author described "These inhibitors did not have substantial effect in OE21ER cells despite adequate biochemical target engagement (fig. 3 d, e, Supplementary Fig. 4)", but as shown in figure 3e, we found Trametinib can significantly inhibit OE21ER cells growth, while Rapamycin also had a substantial effect in OE21ER cell growth in sup. figure 4. The author should explain the reason.

We agree that there were modest effects of the addition of the trametinib or rapamycin in the erlotinib resistant models. We have adjusted the text to state that, "These inhibitors did not have substantial effect in OE21ER cells despite adequate biochemical target engagement". However, we believe that this modest effect does not substantively modify our conclusion. That is, these results do not support the proposition that with EGFR inhibition that one could

wait until acquired resistance and then overcome that resistance with the combination of EGFR blockade with either MEK or mTOR inhibition. In contrast, the uses of upfront combinations that block (or least greatly delay) onset of resistance appear to be a better alternative.

16. In figure 4a, the results lacked DMSO group; the author described d "ERK1/2 reactivation was observed within just 48 hours following continuous exposure to treatment with erlotinib or afatinib, despite a lack of return of the phosphorylation of AKT or EGFR (Fig. 4a)." In fact, as shown in figure 4a, phosphorylation of AKT reactivation was also observed within just 48 hours following continuous exposure to treatment with erlotinib or afatinib. The author should explain the reason.

In figure 4a (now is figure 3a) we show the comparison between baseline and erlotinib-afatinib- treated groups and have a time DMSO control (at left of figure). The reviewer makes an important observation that there is some re-activation of p-AKT in addition to p-ERK and we now note this change in our modified text. A presumable explanation is some degree of up-regulation of an upstream signaling pathway, which leads to strong p-ERK and some degree of p-AKT activation. What most relevant to highlight, however, is the strength of the p-ERK activation and the functional significance of dual blockade of the MAPK/Cell cycle pathway and EGFR.

17. Sup, figure 6 had disordered composition; ERK1/2 reactivation was not observed within 48 hours following continuous exposure to treatment with erlotinib or afatinib in K140 cells, while author described "Other EGFR inhibitor sensitive-lines showed similar ERK reactivation (Supplementary Fig. 6)."The author should explain the reason.

We have extended our time course to assess signaling in the KYSE 140 cells as such rebound was not noted at 24 hours (as the reviewer states). This Figure is now moved to Supplemental Figure 8 and shows the p-ERK rebound at 48 and 72 hours post drug.

18. Sup. figure 7 did not mark the columns to represent different inhibitors.

We have now marked the columns in the revised manuscript and hope this enhances the clarity of the figure.

19. In figure 4c,d,e, why occurred different treated times for the same inhibitors. How many concentrations were chose for each inhibitors?

We have now showed the concentration of each inhibitor in these figures. With our experiments, we evaluated the cell cycle state at 48 hours after the dosing given our expectation that this phenotype would be more rapidly apparent. We performed our apoptosis measurements at 72 hours after dosing to enable time for clearer apoptosis induction.

20. Sup, figure 8 had disordered composition; it did not mark the columns to represent different inhibitors, and also lacked significantly statistical analysis.

We have now marked the columns and added the markings to demonstrate the statistical significance in these figures (as detailed in the legend).

21. In sup. figure 9, I did not see the results about Rb phosphorylation;

Rb phosphorylation has been added in this figure, which in the revision is now supplementary fig 12.

22. I did not find the "figure 6" in the text.

We now have renamed the Figure to correct this omission.

23. The statistical methods described here were not integrated.

We thank the reviewer for this suggestion and have have now added additional details to the statistical sections of the methods.

24. I urge the authors to invite some native English-speaking scientists to revise the manuscript.

We have invited some native English-speaker to revise the manuscript.

Reviewer#2 (cell cycle, check points and cancer)

The authors describe progress in identify drug combinations for use in ESCC tumor cell lines bearing EGFR amplifications, a common driver of this disease. Although some such tumors respond to EGFR inhibitor therapy initially, resistance develops, and others are de novo resistant. The authors show that downstream signaling can be reactivated, as in numerous other tumor types, and MEK inhibition, for example, can be helpful. However, once resistance develops, it often involves EMT, and such cells are poorly responsive even to combination therapies. The authors go on to show that EGFR amplified ESCC often contain RB pathway mutations, almost always involving loss of p16 or amplification of cyclin D or cdk6, consistent with numerous previous studies of OSCC. Interestingly, cdk4/6 inhibitors can somewhat augment EGFR inhibitors in cell culture with arrest and apoptosis endpoints, but have a more robust effect on preventing colony outgrowth when given in combination, and are very effective at preventing xenograft growth of one cell line. The take home message is that combinatorial therapy with cdk4/6 inhibition could prevent outgrowth of resistant cells, and thus the generation of highly intractable, EMT cells. Overall the study is interesting and supportive of the concept of a clinical trial for this combination. On the downside, few new concepts are uncovered and the efficacy of the combination is not surprising in light of other studies, including the authors' reference 25, interrogation PI3K + cdk4/6 inhibition in breast cancer. However, given the potential clinical implications of this study, it is nevertheless likely to be of significant impact. The following issues are important to address:

We thank the reviewer for providing helpful comments and acknowledging the clear clinical relevance of our study. Specific concerns and comments are addressed below.

1. The response of OE21 and KYSE140 to palbociclib is quite different molecularly, but (apparently) not in terms of cell outgrowth. It is not clear if this reflects differences in response to palbociclib, because contrary to the text, no data for RB phosphorylation are presented in figure S9, so this cannot be assessed. This critical point needs to be addressed before a full evaluation is possible. On a more minor note, it is not clear if Figure 4f and 5e present the two cell lines as this is not labeled, but this is assumed to be the case.

We thank the reviewer for pointing out this oversight in the original submission. We have added the phospho-RB and total-RB immunoblots in what was supplementary figure 9 (now suppelemental figure 12). As shown, erlotinib could significantly inhibit the phospho-RB and tot-RB, and palbociclib could partially inhibit phospho-RB but not total RB. We also clarify the cell lines used in the experiments in Figure 4f (now Figure 3f) and 5e (now Figure 4e).

2. It is a bit concerning that only one of the two cell lines is tested in vivo, given the dramatic response in vivo to dual inhibition that seems disproportionate to the proliferation and

apoptosis response. Is a strong in vivo response also seen with KYSE140? Fundamentally, it is important to know that more than one cell line responds in this way, and ideally more than two, if molecular responses are different.

We agree with the reviewer and have now used a second cell line KYSE140 in our in vivo study. Specifically, we repeated the studies testing the combination of combination of Erlotinib and Palbociclib in KYSE140 cell lines grown as xenografts in nude mice. As shown in the new supplementary figure 16, only the combination of erlotinib plus palbociclib but not monotherapy (erlotinib or palbociclib) showed significant in vivo tumor inhibition effect in KYSE140 xenograft experiments. As there are other prior studies which have evaluated the possible efficacy of EGFR/MEK combinations in other cancer types we did not believe it necessary to retest the KYSE140 xenografts with erlotinib combined with trametinib.

3. Pathologic analysis of tumors in figure 10 should be included to assess if combinations are augments apoptosis, arrest, both, etc.

We have now added the pathologic analysis of tumors in a new supplementary figure 15. In this new figure, we now evaluate the proliferation marker (Ki67), and apoptosis markers (caspase3) and data were shown in supplementary fig 15. IHC analysis demonstrated that, slightly decrease of Ki67 and increase of caspase 3 expression were seen in erlotinib and trametinib single agent. Palbociclib monotherapy showed modestly decreased of the ki67 expression but not strong induction of caspase 3. E+T showed more caspase 3 expression but similar ki67 expression compared with vehicle. E+P was strongly correlated with decreased tumor proliferation by Ki67 staining and modest caspase 3 induction.

minor point - RB pathway mutations are described in supplemental table 2, not 1, as called out in the text.

We thank the reviewer for pointing out this error. We have fixed this issue in the manuscript since the first submission.

Reviewer #3 (Remarks to the Author):

In this study the authors addressed the mechanisms of resistance of esophageal squamous cell carcinoma (ESCC) cell lines to EGFR tyrosine kinase inhibitors (TKI). The authors demonstrate that epithelial mesenchymal transition (EMT) is involved in the acquired resistance to these drugs. They also show that the combination of EGFR TKIs with either MEK or CDK4/6 inhibitors can delay the occurrence of resistance and has a significant effect on the in vivo growth of EGFR TKI sensitive ESCC cells.

Overall, this is an interesting paper providing novel information on the possibility to develop target therapy in ESCC. However, the study also has several limitations and points to be clarified:

We thank the reviewer for these helpful comments and we have now added more important resistant mechanism details to the experimental information as well as new data to support the genotype/phenotype relationship. Specific concerns and comments are addressed below:

1) The authors found that EMT is involved in the acquired and intrinsic resistance to EGFR TKI (at least in some cell lines), but they do not follow up with these data. Rather than trying to identify the molecular mechanism involved in this phenomenon and potential targets for therapeutic intervention, the authors focused on the combination of EGFR TKI and MEK inhibitors to prevent the resistance. I did not find very logical way in which they decided to proceed, and the paper will be improved by additional characterization of resistant cells. In this regard, it will be better to generate additional resistant clones from different cell lines i.e. starting from a different genetic background.

The reviewer makes an important suggestion here. A first modification of our revised submission is to greatly enhance our characterization of phenotype of the resistant cells. We more deeply assessed the changes accompanying EMT in the OE21 model. As we show in new data, we found that not only the mesenchymal markers (N-cadherin and vimentin) are upregulated in erlotinib- and afatinib-resistant OE21 cells. We show now that resistant populations lose expression of SOX2, a squamous lineage transcription factor, and unregulate expression of the AXL tyrosine kinase receptor. The AXL finding is notable as AXL activity has been implicated as an etiology of resistance to EGFR-targeted therapy in lung cancer and associated with EMT transition. These data are in Figure 2d. We have extended these observations by evaluating the 2 clearly de novo EGFR inhibitor resistance cell lines TE8 and KYSE520, which also show AXL expression and loss of SOX2 consistent with their other mesenchymal markers. Beyond these cell lines, we also exposed the KYSE140 line to long term treatment (2 months) to increasing doses of erlotinib and afatinib. Although we did not find observe the same change in mesenchymal n-cadherin expression with induction of

resistance (as was seen with OE21), we observed AXL upregulated in resistance KYSE 140 lines (as shown in Supplementary fig 4).

Beyond these studies, we performed further analysis to assess the hypothesis that AXL contributes to the intrinsic and acquired resistance to EGFR inhibition. We first evaluated the inhibitor R428, which inhibits AXL and asked if this agent either showed efficacy or enhanced the antiproliferative activity EGFR of blockade in EMT-associated intrinsic and acquired resistant cell lines (Supplementary fig 6 a-e). The results were mixed with clear reduction of proliferation with R428 at a dose of 1uM. However, with this dose, we did not observe either substantive synergy with EGFR blockade or an effect upon enhancing effective blockade of key pathways downstream of EGFR. Therefore to get a better assessment of whether AXL is clearly a driver of resistance, we also evaluated genetic targeting of this kinase with siRNA. With silencing of AXL, we did not observe any re-sensitization of these EMT-status cells to EGFR inhibition or any enhanced blockade of EGFR downstream pathways (Supplementary fig 7 a-e). Those findings suggesting that AXL inhibition may not be an effective means to sensitize these tumors to EGFR blockade in the setting of EMT. More broadly, these data speak to the clear challenges associated with effective therapies of tumors with a more mesenchymal state.

Getting back to the other point of the reviewer, we wish to address the question regarding the overarching logic of the paper. We completely agree with the reviewer regarding the importance of further study of the mechanisms of resistance. However, we also recognize that treating after onset of resistance can be therapeutically challenging (except in cases with a simple monogenic etiology of resistance such as a secondary EGFR-mutation in lung cancer). Therefore, we have opted to focus in this manuscript upon the possible strategies that could potentiate the initial efficacy of EGFR blockade. Especially since there are such possibilities with existing FDA-approved drugs, we hope these results will spur testing of therapies that can immediately have impact upon patients.

2) The authors show in fig 4a a rebound of ERK1/2 activation following continuous exposure to erlotinib. This experiment needs to be better explained: did the authors change the medium during the experiment? Was erlotinib added during the experiment or cells were treated only once? These factors can make a significant difference on the results of this type of experiments.

We agree with this importance of experimental conditions with such experiments. We performed these experiments in two distinct manners. First, we did the experiment without media change (with a single treatment). We also performed the study with media changes where fresh media/fresh drug were replenished daily during the experiment. In both situations, we observed the same rebound of ERK1/2 after 48 and 72 hrs (shown below). We have not yet added these data to the manuscript, but would readily do so if the reviewer believes it would enhance the paper.

3) The combination of trametinib and erlotinib does not seem much more effective as compared with erlotinib alone in fig 4c. This is also true for many other combination experiments showed in the paper. If the authors want to claim that a synergism occurs with these combinations, they need to perform isobolographic analysis of drug combinations.

We agree with the reviewer and have now added synergistic test for erlotinib plus trametinib or palbociclib in both OE21 and KYSE140 cells (data showed in supplementary fig 9, 13). Combinations erlotinib and trametinib worked synergistically in OE21 cells (CI value: 0.52962, R value: 0.98760) but not KYSE140 (CI value: 1.43592, R: 0.95953). Erlotinib and

palbociclib showed synergistic activity in both cells (OE21 CI value: 0.41060, R value: 0.99613; KYSE140 CI value: 0.66411, R value: 0.99830).

4) The combinations of erlotinib with either trametinib or palbociclib seem to delay the occurrence of resistant clones. However, the authors should clarify whether resistant clones do develop at a certain time. In addition, it would be important to know whether the resistant clones undergo EMT as a major mechanism of resistance.

We have now performed additional experiments, which are detailed in the text and shown in new Supplementary fig 10. We performed treatment studies in vitro with combinations of erlotinib with trametinib or with palbociclib in both the OE21 and KYSE140 cell lines. In both cell lines, with the combination of erlotinib and palbociclib, we failed to observe any outgrowth of resistant cells in either cell lines. With the combination of the erlotinib and trametinib, however, we observed resistant cells which grew out in the OE21 cells but not the KYSE140. We evaluated these EGFR/MEK inhibitor resistance cells (as shown in Supplementary Figure 10), and again found markers of EMT in these cells. We still do not know if such resistant cells are pre-existing as a minor sub-population or emerge in response to therapy.

5) In the in vivo experiments, the combinations of erlotinib with either trametinib or palbociclib seem to be more effective as compared with a single drug. However, is the difference statistically significant?

Combinations showed significantly better outcomes compared with monotherapy. Statistical testing is included in these figures.

6) The authors should analyze tumors from mice treated with single agent in order to assess whether EMT is induced also in vivo.

We also add the pathologic analysis of tumors in supplementary figure 15. We tried to grade IHC for Ecadherin, Ncadherin and Vimentin, but could not find substantial difference between the erlotinib and vehicle groups. The discrepancy between in vitro and in vivo study may be because we use consistent dose of erlotinib during the in vivo experiment, while we did increase the dose gradually when we culture the resistant clones in vitro.

7) The manuscript should be checked for a number of minor errors affecting its quality. For

example, in the text the author refer to supplementary table 1 for the genetic features of cell lines that are described in supplementary table 2. In addition, the author refer to Rb phosphorylation shown in Supplementary fig 9. The version that I received of this figure does not contain blots for Rb phosphorylation.

We thank the reviewer for point these errors. We have now fixed those errors in the revised manuscript.

REVIEWERS' COMMENTS:

Reviewer #2 (Remarks to the Author):

The authors have satisfied my concerns. While the molecular novelty of the study is not at the highest level, the translational/clinical implications of this work, particularly in the ability to impair the emergence of resistance, are sufficient to be of interest to the readership of this journal.

Reviewer #3 (Remarks to the Author):

The authors have addressed most of my concerns and have added new experimental data. The manuscript is significantly improved.